# Simulation-Based Design of a Silicon SPAD with Dead-Space-Aware Avalanche Region for Picosecond-Resolved Detection

**DOI:** 10.3390/s25196054

**Published:** 2025-10-02

**Authors:** Meng-Jey Youh, Hsin-Liang Chen, Nen-Wen Pu, Mei-Lin Liu, Yu-Pin Chou, Wen-Ken Li, Yi-Ping Chou

**Affiliations:** 1Department of Mechanical Engineering, Ming Chi University of Technology, New Taipei City 243, Taiwan; mjyouh@mail.mcut.edu.tw; 2Research Center for Intelligent Medical Devices, Ming Chi University of Technology, New Taipei City 243, Taiwan; 3Department of Mechanical Engineering, Chang Gung University, Taoyuan 333, Taiwan; 4Department of Semiconductor Engineering, Lunghwa University of Science and Technology, Taoyuan 333, Taiwan; dd044@mail.lhu.edu.tw (H.-L.C.); meilin.liu@mail.lhu.edu.tw (M.-L.L.); 5Department of Electric Engineering, Yuan Ze University, Chung-Li, Taoyuan 320, Taiwan; nwpuccit@saturn.yzu.edu.tw; 6Department of Power Mechanical Engineering, National Tsing Hua University, Hsinchu 300, Taiwan; s113033622@m113.nthu.edu.tw; 7Department of Mechanical Engineering, Chung Yuan Christian University, Taoyuan 320, Taiwan

**Keywords:** single-photon avalanche diode (SPAD), dead-space effect, electric field confinement, COMSOL Multiphysics, guard ring design, avalanche gain stability, picosecond time resolution, silicon photonics

## Abstract

**Highlights:**

**What are the main findings?**

**What is the implication of the main finding?**

**Abstract:**

This study presents a simulation-based design of a silicon single-photon avalanche diode (SPAD) optimized for picosecond-resolved photon detection. Utilizing COMSOL Multiphysics, we implement a dead-space-aware impact ionization model to accurately capture history-dependent avalanche behavior. A guard ring structure and tailored doping profiles are introduced to improve electric field confinement and suppress edge breakdown. Simulation results show that the optimized device achieves a peak electric field of 7 × 10^7^ V/m, a stable gain slope of −0.414, and consistent avalanche triggering across bias voltages. Transient analysis further confirms sub-20 ps response time under −6.5 V bias, validated by a full-width at half-maximum (FWHM) of ~17.8 ps. Compared to conventional structures without guard rings, the proposed design exhibits enhanced breakdown localization, reduced gain sensitivity, and improved timing response. These results highlight the potential of the proposed SPAD for integration into next-generation quantum imaging, time-of-flight LiDAR, and high-speed optical communication systems.

## 1. Introduction

Single-photon avalanche diodes (SPADs) are indispensable devices for applications requiring high temporal precision, such as quantum imaging, time-of-flight (TOF) LiDAR, fluorescence lifetime imaging (FLIM), and optical communication. Their ability to detect single photons with sub-nanosecond timing has enabled broad advances in both fundamental science and practical technology [1,2,3].

Over the past decades, the development of SPADs has been closely tied to advances in device engineering, especially in electric field management. Early works, such as Batdorf (1960) [4], already emphasized the importance of guard rings in suppressing premature edge breakdown in avalanche diodes. Later, Kindt (1999) [5] provided a comprehensive theoretical framework for SPAD operation, including afterpulsing and timing jitter. More recently, Wang et al. (2017) [6] demonstrated floating guard rings as an effective means of enhancing breakdown confinement, while Helleboid et al. (2022) [7] analyzed the temporal response of SPADs in the context of high-speed imaging applications. These studies underscore that guard ring structures are an established design feature, but their optimization remains critical to balancing breakdown localization [8,9,10], gain stability, and temporal resolution.

In this work, we perform a comparative analysis between SPAD structures with and without guard rings to explicitly highlight the confinement effects introduced by this design. Rather than implying that guard rings are absent in conventional SPADs, our intention is to demonstrate, within a controlled modeling framework, how field shaping translates into measurable improvements in avalanche stability and timing resolution.

For simulation, we employed COMSOL Multiphysics to construct a two-dimensional SPAD model incorporating a dead-space-aware impact ionization framework. While TCAD platforms such as Synopsys Sentaurus and Silvaco Atlas are widely established for SPAD design, COMSOL provides a flexible alternative for coupling semiconductor transport with multiphysics analysis. The novelty of our study lies not in the use of COMSOL itself, but in applying a dead-space-aware avalanche model to investigate picosecond-level transient behavior, thereby bridging the gap between device-level physics and timing performance.

By integrating guard ring confinement with dead-space-aware modeling, we show that the proposed design achieves sub-20 ps temporal resolution and stable avalanche gain characteristics, positioning it as a strong candidate for next-generation LiDAR, quantum imaging, and high-speed photonic applications.

## 2. Materials and Methods

Figure 1 illustrates the cross-sectional layout of the simulated SPAD device, including the n^+^, p, and p^+^ silicon layers, the guard ring, the top surface passivation, the lateral field stop oxide, and the outermost STI oxide enclosure. The model was constructed in COMSOL Multiphysics^®^ 6.3 (COMSOL AB, Stockholm, Sweden) using a two-dimensional (2D) cross-sectional geometry. This simplified layout represents a vertical slice through the active region of the SPAD. In practice, the device active area is typically circular or nearly circular; however, the 2D approximation provides computational efficiency while preserving the essential vertical and lateral field-shaping mechanisms.

In the proposed design, the guard ring and the field stop are implemented on opposite sides of the multiplication region with distinct doping profiles. The guard ring is moderately doped to laterally confine the high-field region, suppressing premature edge breakdown, while the field stop oxide serves as an insulating barrier that prevents field crowding at the device periphery. The asymmetry in their configuration allows for evaluating complementary strategies for lateral breakdown suppression and provides insight into the role of selective doping in avalanche localization.

To accurately simulate the physical behavior of the SPAD, the following COMSOL modules and configurations were utilized:Semiconductor Module—Used to define the core device regions including the n^+^, p, and p^+^ regions, as well as the guard ring, enabling the resolution of carrier transport and recombination.Stationary Study—Applied to model the steady-state distribution in the stop region and outer insulating layers, ensuring boundary consistency and electric field convergence.Avalanche Generation with Okuto–Crowell Model—Incorporated specifically in the gain region to simulate the history-dependent nature of impact ionization and to characterize the onset of avalanche multiplication.

For reproducibility, the exact device parameters used in the COMSOL simulations are summarized in Table 1. The guard ring was designed with a lateral width of 0.25 μm and a depth of 0.35 μm. The n^+^ and p^+^ regions were heavily doped to ~1 × 10^20^ cm^−3^, while the central p-type region was lightly doped to ~1 × 10^16^ cm^−3^. The Okuto–Crowell model parameters included critical ionization field values and carrier mobility consistent with standard silicon data. These specifications ensure that the spatial field shaping and avalanche dynamics are fully reproducible for researchers intending to replicate our dead-space-aware modeling framework.

It should be noted that the device cross-section was modeled using simplified rectangular doping profiles without diffusion tails or implantation gradients. This abstraction was intentionally chosen to establish a proof-of-concept framework focused on the influence of dead-space-aware avalanche modeling and guard ring confinement on timing performance. While this approach provides clear physical insight into avalanche localization and transient behavior, it does not fully capture the complexity of real fabrication processes. Future work will extend the model by incorporating Gaussian-like doping distributions and process-based TCAD simulations (e.g., Synopsys Sentaurus, Silvaco Atlas) to further validate the findings under fabrication-realistic conditions.

## 3. Results

### 3.1. Electric Field Distribution

Figure 2 shows the simulated electric field distribution at V_bias_ = −6 V with the guard ring incorporated. The electric field reaches a peak of approximately 7 × 10^7^ V/m at the junction curvature near the multiplication region while remaining well confined to the central core. In contrast, Figure 3 illustrates the case without the guard ring, where the absence of this field-shaping structure results in a flatter and more diffuse distribution, with a reduced peak magnitude (~3.5 × 10^7^ V/m). This indicates weaker confinement and a higher susceptibility to edge-triggered breakdown.

To further evaluate the effectiveness of field control, Figure 4 presents the horizontal electric field profiles at y = −0.7 μm for both device types. The SPAD with a guard ring exhibits a sharply peaked field near the central region, rising to ~4.6 × 10^7^ V/m before rapidly decaying toward the edges. This confirms that the guard ring effectively concentrates the electric field within the multiplication zone while suppressing lateral field diffusion. By comparison, the device without a guard ring shows a relatively uniform field profile of ~4.2 × 10^7^ V/m across the active width, suggesting weaker localization and an increased risk of lateral breakdown extension.

It is important to note that avalanche multiplication in silicon becomes significant once the local electric field exceeds ~3.5 × 10^7^ V/m, which serves as the threshold for impact ionization. The proposed design, achieving a peak field of 7 × 10^7^ V/m, ensures that the multiplication region consistently operates above this critical level while maintaining confinement. This guarantees reliable avalanche triggering in the intended zone without premature breakdown at the periphery. Overall, the inclusion of a guard ring provides several advantages: (i) it enhances both vertical and lateral confinement of the high-field region, thereby reducing susceptibility to premature peripheral breakdown; (ii) it enables faster and more controlled avalanche initiation through stronger field localization; and (iii) it contributes to lower multiplication noise and improved temporal resolution, both essential for precise photon timing and low-jitter operation.

### 3.2. Impact Ionization Behavior Using Okuto-Crowell Model

Figure 5 shows the simulated spatial distribution of the impact ionization rate (*G_ii_*) at V_bias_ = −6 V using the Okuto-Crowell model. The ionization activity is strongly confined to a narrow region beneath the n^+^/p^−^ junction, where the electric field exceeds the critical threshold for avalanche triggering. This localization confirms that the designed multiplication region operates as intended and that avalanche initiation is reliably established in the central core of the device.

Although a weak secondary ionization tail can be observed near the p^+^/p^−^ interface, this effect arises from the simplified symmetric doping profiles in the present model. Importantly, the dominant avalanche activity remains concentrated in the n^+^/p^−^ region, and the parasitic field peak near the p^+^/p^−^ interface contributes negligibly to the overall multiplication process.

The sharp vertical gradient of the ionization rate at the n^+^/p^−^ junction reflects the steep electric field variation in this region, supporting efficient and well-controlled carrier multiplication. These results highlight both the reliability of the dead-space-aware modeling framework and the effectiveness of the guard ring in suppressing premature edge-triggered breakdown. Overall, the analysis confirms that the primary avalanche zone is correctly located at the n^+^/p^−^ junction, ensuring robust timing response and stable device operation.

### 3.3. Gain Linearity and Sensitivity Comparison

To evaluate the effectiveness of the guard ring design, we extracted the total impact ionization rate (*G_ii_*) across a range of reverse biases and plotted the logarithmic transformation (*log*_10_(*G_ii_*)) versus V_bias_. In this context, the gain slope is defined as$$s=\frac{d({log}_{10}(G_{ii}))}{dV}$$
with units of V^−1^. This metric provides a quantitative measure of how sensitively the avalanche multiplication responds to variations in bias voltage. A smaller absolute slope indicates reduced gain sensitivity and thus greater stability against bias fluctuations.

Figure 6 shows the linear regression result for the proposed device with guard ring, exhibiting a slope of −0.414 with R^2^ = 0.999. This relatively flat slope reflects improved gain stability, as the avalanche multiplication changes more gradually with bias. In contrast, the control device without a guard ring demonstrates a significantly steeper slope of −1.092 and a slightly lower R^2^ value. The sharper slope implies that the control device exhibits higher gain noise and less predictable multiplication characteristics under overbias conditions.

These results highlight the important role of the guard ring in stabilizing avalanche behavior. By mitigating excessive gain sensitivity, the guard ring contributes to noise-resilient operation and ensures reliable photon detection performance across varying bias levels.

### 3.4. Avalanche Gain Sensitivity Analysis

The gain sensitivity in avalanche photodiodes is closely related to the slope of the *log*_10_(*G_ii_*) versus V_bias_ curve. A flatter slope indicates a more stable avalanche region, with less exponential sensitivity to bias variation. As seen in Figure 7, the gain slope for the device with guard ring is −0.414, which implies relatively stable gain modulation. In contrast, the control device without guard ring (Figure 7) exhibits a much steeper slope of −1.092, highlighting a sharper dependence on V_bias_.

This difference in slope not only reflects enhanced field confinement but also indicates improved suppression of edge-triggered avalanche multiplication. The presence of the guard ring broadens the stable operating voltage range, effectively reducing the risk of uncontrolled breakdown. This supports the overall design objective of achieving low-noise, timing-stable SPAD operation suitable for high-resolution single-photon detection.

From a device physics perspective, the gain slope behavior serves as an indirect indicator of gain noise performance. A flatter *log*_10_(*G_ii_*) slope corresponds to reduced exponential sensitivity to voltage fluctuations, which implies that the multiplication process is more stable under bias variations. This stabilization helps mitigate the inherent stochasticity of impact ionization events, leading to lower gain noise and improved consistency in photon detection. Although gain noise is not explicitly calculated in this study, the reduced gain slope provides strong evidence that the proposed SPAD design suppresses gain-related noise sources more effectively than the control structure.

### 3.5. Transient Response Simulation

To further validate the picosecond-level timing behavior of the proposed SPAD, a time-dependent simulation was carried out using COMSOL’s Semiconductor Module. A photon-triggered avalanche was mimicked by injecting a single electron–hole pair into a localized circular region of radius 0.05 μm positioned at the center of the multiplication zone. The initial carrier concentration within this region was set to 1 × 10^15^ cm^−3^ (1 × 10^21^ m^−3^) for both electrons and holes. The Time-Dependent Study was configured over 0–100 ps with a temporal resolution of 0.1 ps. Reverse bias voltages of −5.5 V, −6.0 V, and −6.5 V were applied to examine the dependence of avalanche initiation dynamics on electric field strength.

Figure 8 presents the transient terminal currents for the three bias conditions. The results show that higher bias voltages lead to faster avalanche triggering and larger peak currents, whereas lower bias conditions yield slower current buildup and reduced amplitude. This behavior directly reflects the field-driven nature of impact ionization: as the bias increases, carriers gain sufficient energy over shorter mean free paths, thereby raising the probability of ionization events per unit distance. Consequently, avalanche buildup occurs more rapidly, leading to lower latency and improved stability.

To quantify the timing response, we extracted the full width at half maximum (FWHM) of the avalanche transient. At −6.5 V bias, the current transient exhibited an FWHM of ~17.8 ps, confirming the sub-20 ps temporal resolution of the proposed design. This represents a substantial improvement compared to recent reports, such as ~50 ps in a digital SiPM prototype by Rastorguev et al. [11] and ~95 ps in a CMOS-compatible SPAD array by Gulinatti et al. [3]. A comparative summary is provided in Table 2, which highlights the superior timing precision and gain stability achieved in this work.

In addition, prior work by Wang et al. [6] introduced floating guard rings to mitigate edge breakdown, demonstrating the benefits of field confinement. However, their study did not specifically address time-resolved avalanche dynamics. Our design integrates both robust breakdown suppression and direct timing improvements through guard ring–enabled field shaping, making it particularly suitable for applications where picosecond resolution and low jitter are essential, including quantum imaging, LiDAR, and high-speed optical communication.

Overall, these transient simulations confirm that the proposed SPAD achieves controlled avalanche initiation, enhanced temporal resolution, and stable gain behavior, directly attributable to the synergy of dead-space-aware modeling and optimized guard ring engineering.

## 4. Discussion

The simulation results validate the effectiveness of the proposed SPAD structure in achieving strong electric field confinement, stable gain behavior, and rapid avalanche response. The inclusion of a moderately doped guard ring plays a central role in shaping the electric field profile. As demonstrated in Figure 3, Figure 4 and Figure 5, the guard ring successfully localizes the high-field region near the central p–n junction, while significantly suppressing lateral field diffusion and minimizing the risk of edge-triggered breakdown. In contrast, the control structure without a guard ring exhibits a flatter and more dispersed electric field, which could result in unstable multiplication and increased dark count rate (DCR).

The dead-space-aware Okuto–Crowell model further confirms that avalanche triggering is confined to a well-defined region under the junction, with negligible impact ionization activity near the device edges. This level of spatial confinement is essential for consistent carrier multiplication and helps reduce the stochastic variation that contributes to timing jitter. The observed peak impact ionization rate aligns well with the field intensity and demonstrates the predictive capability of the model under various bias conditions.

Moreover, the analysis of gain behavior reveals a critical distinction between the two structures. The SPAD with a guard ring exhibits a flatter gain slope (−0.414), indicating lower sensitivity to bias variations and improved avalanche stability. In contrast, the steeper gain slope (−1.092) observed in the control device suggests higher gain noise and greater susceptibility to overbias effects. This finding implies that spatial field control can directly influence noise performance and device linearity—parameters crucial to applications requiring reliable photon counting and temporal accuracy.

The time-dependent simulations further highlight the benefit of optimized field distribution in achieving fast avalanche onset. Under −6.5 V bias, the proposed structure demonstrates sub-20 ps rise time and higher peak terminal current, illustrating its suitability for high-speed detection applications. These transient results complement the steady-state gain analysis and confirm that the field confinement introduced by the guard ring not only improves breakdown localization but also accelerates the response time of the device. In the revised Figure 8, the transient currents at −5.5 V, −6.0 V, and −6.5 V are replotted using clearly distinguishable colors and markers, ensuring that each bias condition can be easily identified.

An important consideration is the trade-off between guard ring confinement and active area utilization. In our present single-pixel design, the inactive border introduced by the guard ring is estimated to occupy less than 5% of the total active diameter, which represents only a minor compromise compared to the substantial improvement in breakdown suppression and timing precision. It should be noted, however, that for smaller pixel sizes the relative contribution of the guard ring to the inactive area would become more significant, and this scaling effect must be carefully considered in future array-level designs.

Benchmarking against recent literature (Table 2) demonstrates the clear advantage of the proposed design. While prior studies have reported timing resolutions of ~50 ps (Rastorguev et al. [11]) and ~95 ps (Gulinatti et al. [3]), our SPAD achieves a substantially lower transient width of ~17.8 ps. In addition, the stability of the avalanche gain slope distinguishes this work from earlier designs, underscoring the contribution of guard ring–enabled field confinement to both temporal precision and noise suppression.

To avoid possible overestimation, it is important to note that the reported ~17.8 ps timing resolution represents an idealized limit of our COMSOL-based model. The simulations were performed under simplified assumptions that exclude fabrication-induced variations, carrier transport delays, and electronic RC parasitics. In practical implementations, the effective time resolution is expected to be higher. The present value therefore should be interpreted as a theoretical lower bound rather than an immediately achievable experimental performance.

For context, Rastorguev et al. (2025) [11] reported a timing resolution of ~50 ps in a digital SiPM prototype, while Gulinatti et al. (2020) [3] achieved ~95 ps in a CMOS-compatible SPAD array. Compared with these representative results, our simulated design demonstrates a substantial improvement under idealized conditions, indicating the potential of guard-ring-enabled field confinement combined with dead-space-aware avalanche modeling to advance temporal precision. The novelty of this work lies not in the absolute ps-level number itself, but in demonstrating that geometric confinement and history-dependent avalanche physics can be co-optimized to simultaneously enhance gain stability (slope −0.414) and transient resolution. This dual improvement highlights a promising pathway toward low-jitter SPADs suitable for LiDAR, quantum imaging, and high-speed optical communication applications.

Looking ahead, the implications for array-level integration must be considered. When implemented in multi-pixel SPAD arrays, doped guard rings may influence pixel-to-pixel crosstalk and limit the minimum achievable pitch. Previous studies, such as Wang et al. (2017) [6], have investigated the role of floating guard rings in optimizing isolation, suggesting that guard ring width is a key factor in balancing breakdown suppression and fill factor [6,12]. Although our present work focuses on a single-pixel proof-of-concept design, future extensions will explore the trade-offs between guard ring dimensions, pixel pitch, and crosstalk in array configurations.

Despite these promising results, it is essential to acknowledge potential fabrication challenges and the gap between simulation and experiment. Precise control of doping gradients is critical to maintaining the designed electric field distribution, and deviations could increase DCR or shift the avalanche zone. Lithographic alignment of the guard ring also poses a challenge, as misalignment may weaken lateral confinement. Furthermore, junction curvature variations and process-induced defects could degrade timing and noise performance in fabricated devices. These factors highlight the need for careful process optimization and experimental validation, which we plan to pursue in future work.

Overall, the combined results emphasize that both geometric and doping optimizations contribute synergistically to SPAD performance. By integrating dead-space-aware modeling and transient simulation, this study provides a comprehensive framework for designing SPADs with enhanced gain stability and picosecond-level timing resolution. The findings underscore the importance of field-engineered structures in achieving the performance requirements of advanced time-resolved applications, including LiDAR, quantum imaging, and high-speed optical communication [13].

## 5. Conclusions

This study demonstrates a comprehensive design and simulation framework for a silicon SPAD optimized for picosecond-resolved detection using COMSOL Multiphysics. The incorporation of a guard ring and optimized doping profiles significantly enhanced electric field confinement in the central avalanche region, while effectively reducing lateral field extension that could lead to premature edge breakdown.

Comparative electric field analysis revealed that the guard ring design produces a pronounced central field peak with rapid fall-off towards the edges, indicating strong spatial localization. In contrast, the structure without a guard ring displayed a flatter and less confined field distribution, which may compromise gain control and breakdown reliability.

Avalanche gain simulations further confirmed that the proposed SPAD structure exhibits a flatter gain slope, indicative of improved gain stability and lower multiplication noise. Time-dependent transient analyses under varying bias conditions demonstrated that stronger reverse bias voltages facilitate faster avalanche initiation and higher terminal current peaks. The device biased at −6.5 V exhibited the most rapid and intense avalanche response, while lower-bias conditions showed delayed turn-on behavior.

Overall, the proposed SPAD design achieves robust spatial field confinement, stable avalanche multiplication, and excellent timing response. These characteristics make it a strong candidate for integration into advanced low-jitter detection systems targeting applications in time-of-flight LiDAR, quantum imaging, and high-speed optical communication.

It should be emphasized, however, that the reported ~17.8 ps timing resolution represents a theoretical lower bound derived from idealized simulations. Practical implementations will inevitably exhibit higher jitter due to fabrication tolerances, carrier transport dynamics, and electronic circuit limitations. This clarification underscores that while the present results highlight the potential of guard-ring-enabled field confinement combined with dead-space-aware avalanche modeling, further experimental validation is essential to bridge the gap between theoretical prediction and real-world performance.

## Figures and Tables

**Figure 1 sensors-25-06054-f001:**
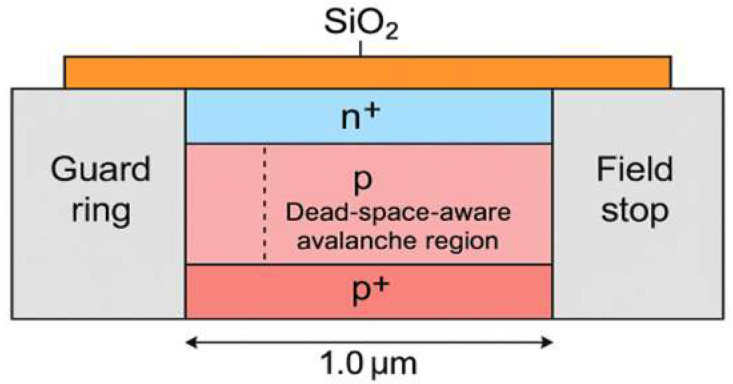
Cross-sectional 2D geometry of the simulated SPAD structure. The model represents a vertical slice through the device, while the actual SPAD active area is typically circular in practice. The structure includes n^+^, p, and p^+^ silicon regions, a moderately doped guard ring on one side, a lateral field stop oxide on the opposite side, top passivation, and an outer STI oxide enclosure. The guard ring provides lateral confinement of the high-field region to suppress premature edge breakdown, whereas the field stop oxide prevents peripheral field crowding. This asymmetric configuration enables evaluation of complementary strategies for electric field shaping and avalanche localization.

**Figure 2 sensors-25-06054-f002:**
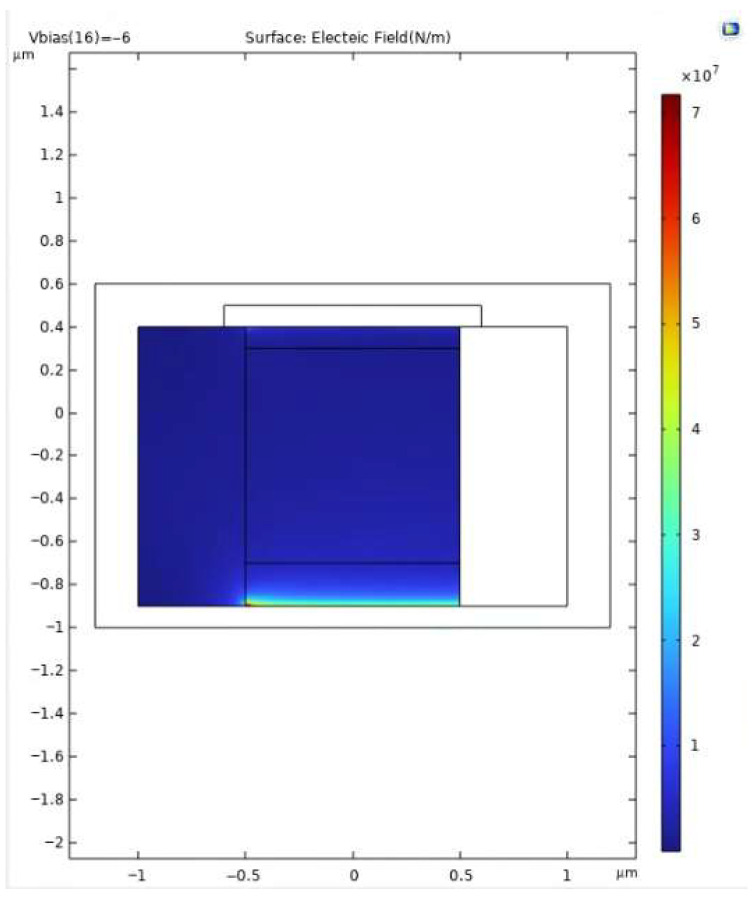
Simulated electric field distribution at V_bias_ = −6 V for the SPAD structure with guard ring. The high-field region is well-confined near the central p–n junction, with the guard ring effectively suppressing lateral field spread and minimizing edge breakdown risk.

**Figure 3 sensors-25-06054-f003:**
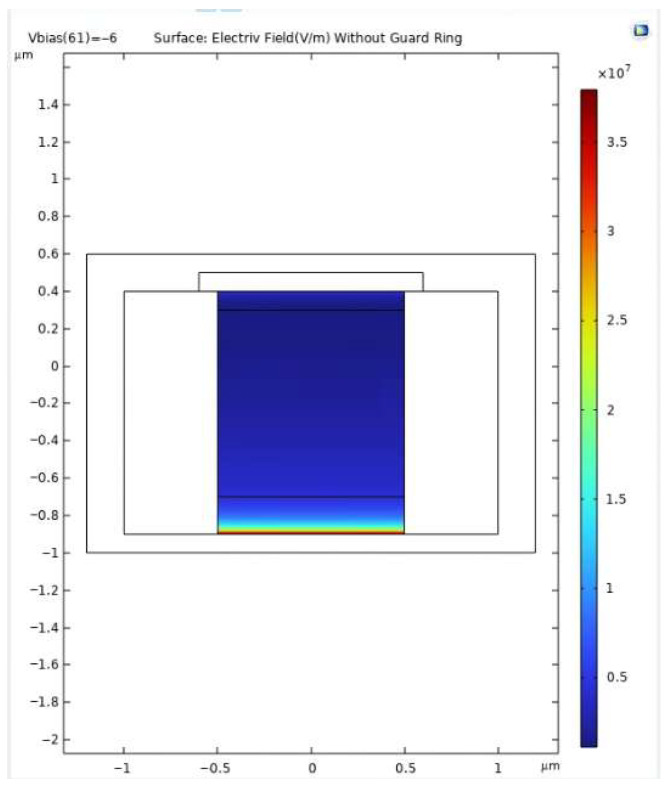
Electric field distribution at V_bias_ = −6 V for the SPAD structure without guard ring. The field is less confined and extends laterally, increasing the risk of premature edge breakdown.

**Figure 4 sensors-25-06054-f004:**
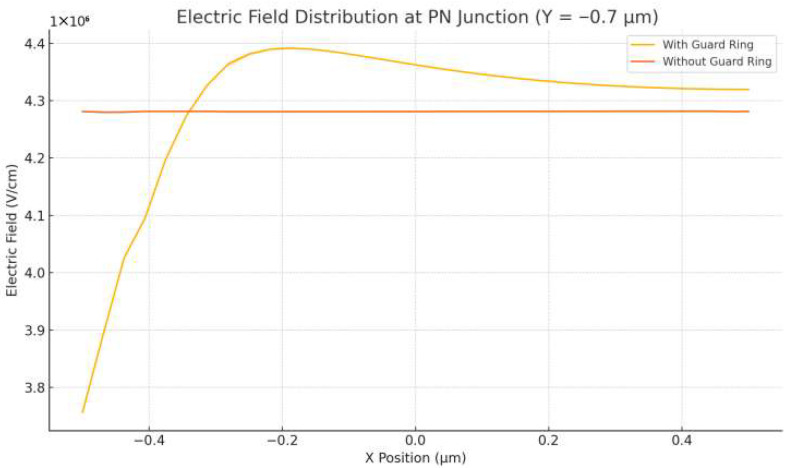
Horizontal cutline of the electric-field magnitude at y = −0.7 μm for SPADs with and without guard rings at V_bias_ = −6 V. The guard-ring device shows a sharp central peak (∼4.6 × 10^7^ V/m) with rapid lateral decay, while the control device exhibits a flatter profile (∼4.2 × 10^7^ V/m) across the width, indicating weaker confinement and higher risk of edge breakdown.

**Figure 5 sensors-25-06054-f005:**
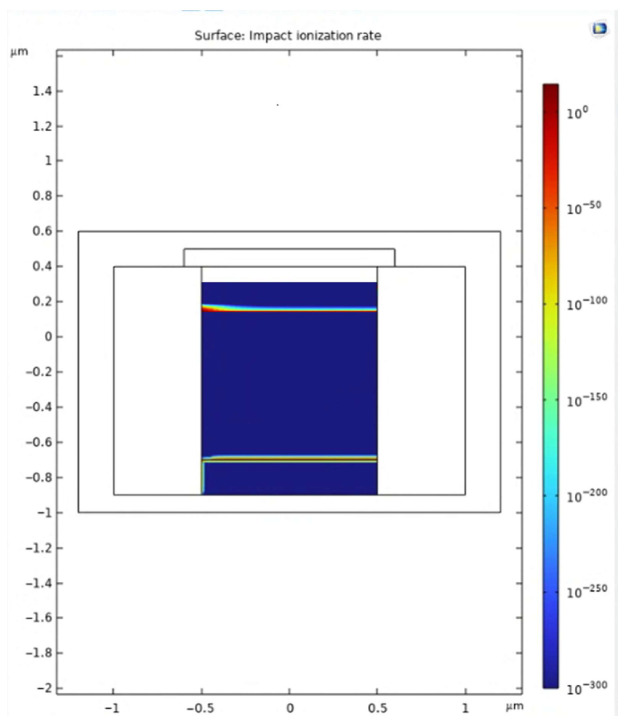
Simulated spatial distribution of the impact ionization rate (*G_ii_*) at V_bias_ = −6.0 V using the Okuto–Crowell model. The avalanche activity is strongly localized beneath the n^+^/p^−^ junction, confirming that this region serves as the primary multiplication zone. Only negligible secondary ionization is observed near the p^+^/p^−^ interface. The logarithmic scale highlights the strong confinement of avalanche triggering within the designed high-field region.

**Figure 6 sensors-25-06054-f006:**
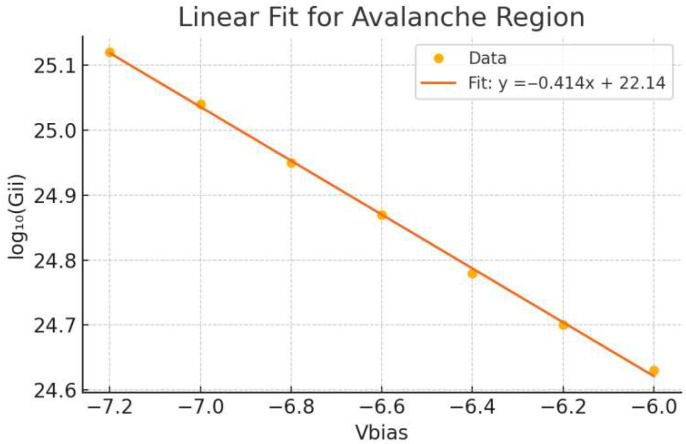
Simulated spatial distribution of the impact ionization rate *G_ii_* at V_bias_ = −6 V using the Okuto-Crowell model. The avalanche activity is well confined to the n^+^/p^−^ junction region, confirming that the main multiplication zone is localized as designed, with negligible ionization near the p^+^/p^−^ interface.

**Figure 7 sensors-25-06054-f007:**
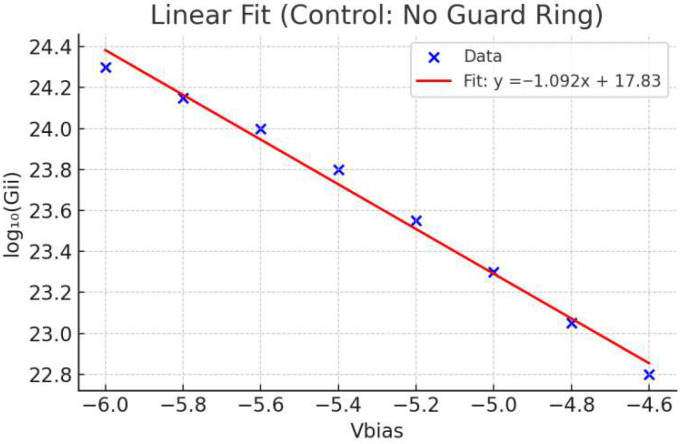
Logarithmic plots of total impact ionization rate *log*_10_(*G_ii_*) versus reverse bias voltage for SPADs with and without guard rings. Linear regression yields slopes of −0.414 V^−1^ (with guard ring) and −1.092 V^−1^ (without guard ring), indicating that the guard ring reduces bias sensitivity and stabilizes avalanche gain.

**Figure 8 sensors-25-06054-f008:**
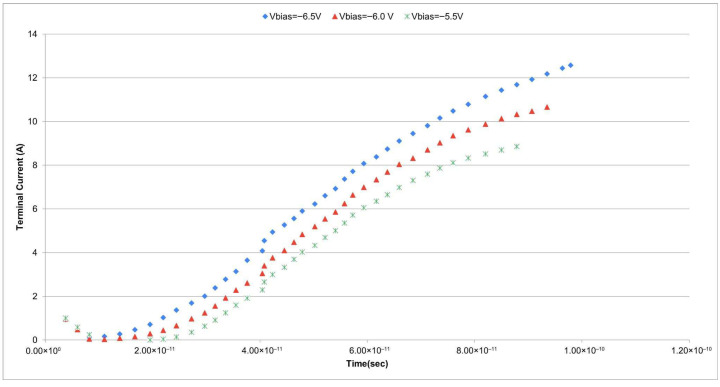
Transient terminal current responses at reverse biases of −5.5 V (red), −6.0 V (green), and −6.5 V (blue). Higher bias accelerates avalanche initiation and increases the peak current, with the −6.5 V case showing the fastest rise and largest amplitude. Distinct colors and a legend are used for clarity.

**Table 1 sensors-25-06054-t001:** Key simulation parameters of the proposed SPAD structure. The simplified rectangular doping profiles were adopted to highlight the effect of guard ring confinement and dead-space-aware avalanche modeling. Values represent the baseline conditions used in COMSOL simulations.

Parameter	Value	Description
Guard ring width	0.25 μm	Lateral confinement dimension
Guard ring depth	0.35 μm	Vertical confinement depth
n^+^/p^+^ doping	~1 × 10^20^ cm^−3^	Contact/high-field regions
p-region doping	~1 × 10^16^ cm^−3^	Avalanche multiplication region
Carrier mobility	1350 cm^2^/V·s (e^−^), 480 cm^2^/V·s (h^+^)	Silicon at 300 K
Critical field (E_a_)	~3.5 × 10^7^ V/m	Avalanche ionization threshold

**Table 2 sensors-25-06054-t002:** Comparison of the proposed SPAD design against recent literature. The inclusion of a guard ring and dead-space-aware modeling enables the proposed device to achieve a timing resolution of ~17.8 ps, significantly outperforming previously reported SPADs (~50–95 ps). In addition, the design demonstrates stable avalanche gain slope (−0.414), highlighting improved robustness in temporal and gain characteristics compared to prior works.

Design/Year	Technology	Reported Timing Resolution	Gain Stability/Slope	Reference
Rastorguev et al. (2025) [11]	Digital SiPM prototype	~50 ps	Not analyzed	IEEE J. Quantum Electron. [11]
Gulinatti et al. (2020) [3]	CMOS SPAD array	~95 ps	Not analyzed	IEEE JSTQE [3]
Wang et al. (2017) [6]	SPAD with floating guard ring	Not specified (focus on breakdown suppression)	Improved edge confinement	IEEE Electron Device Lett. [6]
This work (2025)	COMSOL-based Si SPAD with guard ring	~17.8 ps (FWHM)	Stable slope −0.414	-

## Data Availability

Data supporting the findings of this study are available from the corresponding author upon reasonable request.

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
