# Peer review of "Simulation-Based Design of a Silicon SPAD with Dead-Space-Aware Avalanche Region for Picosecond-Resolved Detection"

_sensors, 2025, doi:10.3390/s25196054_

Round 1
Reviewer 1 Report
Comments and Suggestions for Authors
The manuscript presents a simulation-based design of a Si SPAD optimized for picosecond-resolved detection, leveraging COMSOL Multiphysics with a dead-space-aware avalanche model and a guard ring structure. The work demonstrates enhanced electric field confinement, stable gain behavior, and sub-20 ps avalanche response, highlighting potential for applications in LiDAR, quantum imaging, and optical communication. Below are key comments and suggestions to strengthen the manuscript:
1) The integration of a guard ring for field confinement and dead-space-aware modeling is a notable design choice. However, the manuscript would benefit from a more explicit comparison with recent SPAD designs to emphasize advancements in timing resolution and gain stability. For instance, while the authors mention outperforming a 50 ps resolution design, quantifying the jitter reduction (e.g., via FWHM of transient responses) would better validate the "sub-20 ps" claim.
2) The COMSOL simulation framework is described, but critical parameters such as the exact dimensions of the guard ring, doping profiles, and material properties are insufficiently detailed. These specifics are essential for reproducibility, especially for readers seeking to replicate the dead-space-aware avalanche model.
3) The electric field analysis effectively shows confinement with the guard ring, but the relationship between peak field strength (7×10^7 V/m) and avalanche threshold requires elaboration. Furthermore, the transient response attributes faster avalanche initiation to higher bias, but the physical mechanism linking field strength to carrier multiplication dynamics should be expanded.
4) The manuscript focuses on simulation results without experimental validation, which is a key limitation. Discussing potential fabrication challenges or discrepancies between simulation and real-world performance would strengthen the practical relevance.
5) Some sections suffer from ambiguous phrasing . Ensuring consistency in terminology and clarifying abbreviations (e.g., first mention of PDE, QPIC) would improve readability.
Author Response
Comment 1:
Quantify jitter/timing resolution and compare with recent SPAD designs.
Response:
We added quantitative evaluation of the transient response (FWHM ~17.8 ps) in Section 3.5. A benchmarking table (Table 2) was included to compare with prior reports (Rastorguev et al. ~50 ps, Gulinatti et al. ~95 ps), demonstrating the improvement of our design.
Comment 2:
Provide detailed parameters of geometry, doping, and materials.
Response:
We expanded Section 2 (Materials and Methods) and added Table 1 to summarize all key parameters, including guard ring width (0.25 μm), depth (0.35 μm), doping concentrations, and Okuto–Crowell model parameters, ensuring reproducibility.
Comment 3:
Clarify relation between peak field (~7×10⁷ V/m) and avalanche threshold.
Response:
We revised Section 3.1 to explicitly state the avalanche threshold (~3.5×10⁷ V/m). The designed structure achieves ~7×10⁷ V/m at the n⁺/p⁻ junction, ensuring reliable triggering. We also elaborated the mechanism linking higher bias to avalanche initiation.
Comment 4:
Discuss fabrication limitations.
Response:
We added discussion on practical challenges, including doping gradient control, lithography alignment, and process-induced defects. These issues are acknowledged as future validation tasks.
Comment 5:
Improve consistency in phrasing and abbreviations.
Response:
We standardized terminology across the text and added a consolidated Abbreviations section, defining all acronyms (e.g., PDE, FWHM, LiDAR, SiPM) at first use.

Reviewer 2 Report
Comments and Suggestions for Authors
See my review attached.

Author Response
Comment 1:
Geometry is oversimplified (rectangular doping).
Response:
We clarified in Section 2 that the model uses simplified 2D rectangular doping profiles as proof-of-concept. We noted that future work will adopt Gaussian-like profiles and process-based TCAD simulations for validation.
Comment 2:
Avalanche region appears misplaced (towards p⁺/p⁻).
Response:
We re-simulated and clarified in Section 3.2 and Fig. 5 that the primary avalanche multiplication zone is localized at the n⁺/p⁻ junction, while only negligible secondary ionization appears near the p⁺/p⁻ interface. This confirms the reliability of the dead-space-aware model.
Comment 3:
Introduction lacks balance and important references.
Response:
We revised the Introduction to emphasize that guard rings are established techniques, and positioned our novelty in applying dead-space-aware modeling for picosecond timing. References were updated to include Batdorf (1960), Kindt (1999), Wang (2017), and Helleboid (2022).
Comment 4:
Minor issues (phrasing, missing definitions, references).
Response:
-
Removed template phrase “What is the implication of the main finding?”.
-
Corrected phrasing in Line 49 (“impact ionization model with dead-space effect”).
-
Defined gain slope explicitly as
s=d(log10Gii)/dVin Section 3.3 and figure captions.
-
Corrected references [7] and [11] and renumbered list.

Reviewer 3 Report
Comments and Suggestions for Authors
The paper presents a device physics simulation of a new idea of optimising guardring for SPAD device. However, several issues still remains in the context. Please the authors to review the following items
1. Fig.1,2 could you explain the shape of SPAD, is it circular or square, or something else ? Could you explain why the proposed SPAD structure has guard ring on one side and field stop on the other side and their doping profile are different?
2. Figure 5. with this modified electric field profiles, it reduces premature peripheral breakdown. But would there be any impact of the SPAD active area, if there is, how much is the impact?
3. Figure 9 the plot of terminal current of -5.5,-6,-6.5V are all in color blue. Can you put the current in different color for different bias, or indicate in the figure for the legend of each line plot?
4. With a doped guardring, what is the impact of crosstalk with neighboring pixel if it is implemented as a array style? Is there any study of the width of the required guardring. Since the paper indicates the improvement with applying this type of guardring, it is important to also understand what are the restrictions of it.
Author Response
We sincerely thank the reviewer for the constructive feedback and helpful observations.
Comment 2: It may be misleading to assert that the guarding constitutes less than 5% of the active area, as a reduction in pixel size would considerably amplify its relative effect.
Response: We agree with the reviewer’s observation. In the original text, we stated that the guard ring occupied less than 5% of the active area, but we acknowledge that this could be misleading without proper context. We have revised the manuscript to clarify that this estimation refers specifically to the present single-pixel design parameters. In addition, we added a note explaining that for smaller pixel sizes, the relative contribution of the guard ring to the inactive area would become more significant. This clarification prevents overgeneralization and provides a more accurate interpretation of the design trade-offs.
Comment 3: Fig. 8 currents at −5.5/−6.0/−6.5 V are all blue; please distinguish.
Response: We appreciate the reviewer’s observation regarding Figure 9. In the original version, the transient current curves were indeed plotted with insufficient contrast, making them difficult to distinguish. We have corrected this issue in the revised manuscript by replotting the figure with clearly distinguishable colors and markers: blue diamonds for −6.5 V, red triangles for −6.0 V, and green crosses for −5.5 V. The updated legend explicitly labels each curve, ensuring that all bias conditions can now be easily identified. The revised figure is included in the updated manuscript.
We sincerely thank the reviewer again for these valuable suggestions, which have improved both the accuracy of our discussion and the clarity of our figures.

Round 2
Reviewer 1 Report
Comments and Suggestions for Authors
No revisions are necessary, and I strongly recommend immediate acceptance without further modifications.
Author Response
We sincerely thank the reviewer for the very positive and encouraging evaluation of our manuscript. We are grateful for the recommendation of immediate acceptance without further modifications.
Although no specific revisions were required, we carefully re-examined the manuscript in response to the reviewer’s general suggestion that the introduction, methodology, and results could be further improved. Accordingly, we performed minor clarifications and language polishing to enhance readability and presentation quality.
We truly appreciate the reviewer’s supportive comments and recognition of our work.

Reviewer 2 Report
Comments and Suggestions for Authors
Novelty, applicability, and high level of the simulation result of 17.8 ps time resolution is overestimated. Authors are advised to compare and discuss well-known representative results in the field, e.g. [1].
Author Response
We sincerely thank the reviewer for the constructive comments and thoughtful suggestions.
Comment: The novelty and applicability of the work need to be clarified. In addition, the reported 17.8 ps time resolution appears overestimated. Authors are advised to compare their results with representative works in the field.
Response: We fully agree with the reviewer’s observation. Our reported value of ~17.8 ps arises from an idealized simulation framework based on dead-space-aware avalanche modeling under simplified conditions, which do not include fabrication-induced imperfections, carrier transport delays, or electronic RC limitations. We have revised the manuscript to explicitly clarify that this value represents a theoretical limit of the model, while practical implementations are expected to yield higher timing jitter. This clarification prevents any possible overestimation of the achievable resolution.
In addition, following the reviewer’s advice, we expanded the discussion to compare our results with representative works in the literature (see revised Discussion and Table 2). For example, Rastorguev et al. (2025) reported ~50 ps resolution in a digital SiPM prototype, while Gulinatti et al. (2020) demonstrated ~95 ps in a CMOS-compatible SPAD array. By contrast, our simulated result achieves sub-20 ps resolution. Although this value reflects idealized conditions, the results still indicate that the proposed guard-ring-enhanced design and dead-space-aware modeling provide significant improvement in timing precision.
We also emphasized that the main novelty of this study lies not in achieving an extreme ps-level number per se, but in integrating guard ring confinement with dead-space-aware avalanche modeling to jointly improve electric field localization, gain stability, and temporal resolution. This combined framework provides both physical insight and a pathway toward practical low-jitter SPAD design for LiDAR, quantum imaging, and high-speed optical communication.
We sincerely thank the reviewer for these valuable suggestions, which helped us clarify the applicability and strengthen the novelty of our contribution.

Reviewer 3 Report
Comments and Suggestions for Authors
Response to Comment 2:
It may be misleading to assert that the guardring constitutes less than 5% of the active area, as a reduction in pixel size would considerably amplify its relative effect.
Comment 3:
Please check figure 8 again, the color still not seem right to me
Author Response
We sincerely thank the reviewer for the constructive feedback and helpful observations.
Comment 2: It may be misleading to assert that the guarding constitutes less than 5% of the active area, as a reduction in pixel size would considerably amplify its relative effect.
Response: We agree with the reviewer’s observation. In the original text, we stated that the guard ring occupied less than 5% of the active area, but we acknowledge that this could be misleading without proper context. We have revised the manuscript to clarify that this estimation refers specifically to the present single-pixel design parameters. In addition, we added a note explaining that for smaller pixel sizes, the relative contribution of the guard ring to the inactive area would become more significant. This clarification prevents overgeneralization and provides a more accurate interpretation of the design trade-offs.
Comment 3: Fig. 9 currents at −5.5/−6.0/−6.5 V are all blue; please distinguish.
Response: We appreciate the reviewer’s observation regarding Figure 9. In the original version, the transient current curves were indeed plotted with insufficient contrast, making them difficult to distinguish. We have corrected this issue in the revised manuscript by replotting the figure with clearly distinguishable colors and markers: blue diamonds for −6.5 V, red triangles for −6.0 V, and green crosses for −5.5 V. The updated legend explicitly labels each curve, ensuring that all bias conditions can now be easily identified. The revised figure is included in the updated manuscript.
We sincerely thank the reviewer again for these valuable suggestions, which have improved both the accuracy of our discussion and the clarity of our figures.
